# Simultaneous Determination of 12 Preservatives in Pastries Using Gas Chromatography–Mass Spectrometry

**DOI:** 10.3390/foods12203819

**Published:** 2023-10-18

**Authors:** Liyuan Wang, Zhengyan Hu, Jing Chen, Tianjiao Wang, Pinggu Wu, Ying Ying

**Affiliations:** Zhejiang Province Center for Disease Control and Prevention, Hangzhou 310051, China; lywang@cdc.zj.cn (L.W.); zhyhu@cdc.zj.cn (Z.H.); chenj@cdc.zj.cn (J.C.); tjwang@cdc.zj.cn (T.W.); pgwu@cdc.zj.cn (P.W.)

**Keywords:** pastry, preservatives, gas chromatography–mass spectrometry

## Abstract

(1) Background: Preservatives may pose a potential threat to human health. To ensure food safety, this study has devised a method that concurrently detects a dozen preservatives (acetic acid, propionic acid, dehydroacetic acid, benzoic acid, sorbic acid, dimethyl fumarate, methyl parahydroxybenzoate, ethyl parahydroxybenzoate, propyl parahydroxybenzoate, isopropyl parahydroxybenzoate, butyl parahydroxybenzoate, and isobutyl parahydroxybenzoate) in pastry, utilizing gas chromatography–mass spectrometry. (2) Methods: The pastry samples were acidified with hydrochloric acid, extracted with acetonitrile via vortexing, purified by hexane and saturated with sodium chloride solution to remove lipids and impurities, and then concentrated via nitrogen blowing. The method was then quantitatively analyzed using GC-MS with the internal standard method after methanol re-dissolution. (3) Results: The results showed that the content of the 12 preservatives had good linearity within the range of 1.0–50 μg/mL, with correlation coefficients all greater than 0.99. The method detection limit was 0.04–2.00 mg/kg and the quantification limit was 0.12–6.67 mg/kg. The average recovery rates of the samples at three different spiked concentrations of low, medium, and high were 70.18–109.22%, and the relative standard deviations were 1.82–9.79% (n = 6). (4) Conclusions: This method requires a small amount of sample, has high sensitivity, and is simple and fast to operate, making it suitable for the simultaneous determination of 12 preservatives in pastry. This approach contributes to the effective surveillance and regulation of preservative usage in pastries, thereby safeguarding public well-being.

## 1. Introduction

Preservatives, also known as antimicrobial agents, antifungals, or protectants, exert their effects directly or indirectly on microbial proteins, genetic materials, enzyme systems, and the like, disrupting microbial growth, reproduction, and metabolism. Consequently, they effectively decelerate food spoilage, prolonging the shelf life of products, enhancing economic benefits, and mitigating food poisoning caused by microbial proliferation [1,2]. Food preservatives can be classified as bactericides or bacteriostatics based on their modes of action and as chemical or natural preservatives (also known as biological preservatives) based on their composition and sources. Chemical preservatives can be further divided into three types: acidic, ester-type, and inorganic salt preservatives [3,4]. Currently, China has regulated the use of over 30 food chemical preservatives, including benzoic acid and its salts, sorbic acid and its salts, dehydroacetic acid and its sodium salt, and paraben esters [5].

As the variety of food preservatives continues to increase, the phenomenon of the excessive use of food preservatives in the market is not uncommon. In the increasingly intricate market landscape, the rampant misuse of food preservatives is a recurrent issue. This underscores the heightened importance of researching methods to regulate these preservatives effectively. In order to regulate the use of various preservatives, China has formulated the national standard for the use of food additives, GB2760-2014 “National Food Safety Standard Food Additive Use Standards” [6], and a series of detection methods for different preservatives in different food matrix components have been introduced in the current national and industry standards [7]. However, with the proliferation of types of food preservatives, existing detection methods are confronted with numerous challenges. Currently, laboratory methods for detecting preservatives mainly include thin-layer chromatography [8,9], spectrophotometry [10,11], capillary electrophoresis [12,13], high-performance liquid chromatography [14,15,16,17,18], gas chromatography [19,20,21], gas chromatography–mass spectrometry [22,23,24], liquid chromatography–mass spectrometry [25,26,27], and ion chromatography [28,29]. Nevertheless, as the demands of food safety monitoring escalate, encompassing broader spectrums, and detection tasks become increasingly monumental, there is an urgent need for a high-throughput detection method that is simultaneously efficient, sensitive, precise, and swift in detecting a multitude of preservatives.

To address this issue, this study focuses on establishing an efficient, sensitive, accurate, and rapid gas chromatography–mass spectrometry (GC-MS) method with internal standards. This method aims to simultaneously quantify various preservatives in pastry products. This research not only provides a crucial reference for food safety regulatory authorities but also underscores the purpose of studying food preservatives: to standardize food production and ensure the public’s dietary safety.

## 2. Materials and Methods

### 2.1. Instruments and Reagents

Gas chromatography–triple-quadrupole mass spectrometry (Agilent Technologies, Santa Clara, CA, USA); a vortex oscillator (Shanghai Jingke, Shanghai, China); a constant-temperature water bath (Shanghai Belen Instrument Equipment Co., Ltd., Shanghai, China); an ultrasonic generator (Jiangsu Kunshan Ultrasonic Instrument Co., Ltd., Kunshan, China); an analytical balance (Jinan Bohang Biotechnology Co., Ltd., Jinan, China, weighing range 0–220 g, precision 0.0001 g); a grinding machine (Germany IKA, IKA Instrument Equipment Co., Ltd., Berlin, Germany); a nitrogen blower (Organomation, Berlin, MA, USA); and a centrifuge (Beckman, Brea, CA, USA) were used. Acetonitrile (CH_3_CN): chromatographically pure; ethanol (C_2_H_5_OH): chromatographically pure; hexane (C_6_H14): chromatographically pure; hydrochloric acid (HCl): analytical grade; and methanol (CH_2_OH): chromatographically pure were used. All standard substances, including methyl paraben, ethyl paraben, propyl paraben, isopropyl paraben, butyl paraben, isobutyl paraben, dimethyl fumarate, acetic acid, propionic acid, dehydroacetic acid, benzoic acid, and sorbic acid, were purchased from the National Institute of Metrology of China with a purity of ≥99.0%.

### 2.2. Chromatography–Mass Spectrometry Conditions

#### 2.2.1. Chromatography Conditions

A DB-FFAP capillary column (30 m × 250 μm × 0.25 μm) was used with a temperature program of 60 °C for 1 min, followed by an increase of 15 °C/min to 220 °C, then a further increase of 5 °C/min to 245 °C and held for 10 min. The injection port temperature was set at 300 °C, while high-purity helium was used as the carrier gas with a flow rate of 1.0 mL/min. The split ratio was 7:1, and the injection volume was 1.0 μL.

#### 2.2.2. Mass Spectrometry Conditions

Electron ionization (EI) was used as the ion source with a source temperature of 230 °C. The quadrupole temperature was set at 150 °C, and the transfer line temperature was 280 °C. The electron energy was set at 70 eV, and the solvent delay was set at 4 min. The selected ion monitoring (SIM) mode was used.

### 2.3. Preparation of Solutions

Preparation of standard solution: First, 0.1000 g of each standard compound was accurately weighed, including methyl p-hydroxybenzoate, ethyl p-hydroxybenzoate, propyl p-hydroxybenzoate, isopropyl p-hydroxybenzoate, butyl p-hydroxybenzoate, isobutyl p-hydroxybenzoate, dimethyl fumarate, acetic acid, propionic acid, pyruvic acid, benzoic acid, and shikimic acid, and they were dissolved in 10 mL volumetric flasks with methanol. The volume was made up to the mark with methanol to obtain standard stock solutions with a concentration of 10 mg/mL. They were stored below 4 °C. Before use, they were diluted them with methanol to prepare standard working solutions with different concentrations. Then, 10 mg of accurately weighed D4-ethyl p-hydroxybenzoate was placed in a 10 mL volumetric flask and dissolved with methanol to make up the volume to the mark, to obtain an internal standard solution with a concentration of 1 mg/mL.

### 2.4. Preparation of Standard Curve

Next, 1 mL of each standard stock solution with a concentration of 10 mg/mL was accurately pipetted into a 10 mL volumetric flask and diluted with methanol to make up the volume to the mark, to obtain standard intermediate solutions with a concentration of 1 mg/mL. They were diluted with methanol step by step to prepare standard series solutions for each target compound with concentrations, as shown in Table 1. The concentration of the internal standard D4-ethyl p-hydroxybenzoate in the standard series solutions was 50 μg/mL.

Then, 1 μL of the above standard series solutions were injected, and the standard curve with peak area was plotted as the ordinate and concentration as the abscissa.

### 2.5. Sample Preparation

Next, 1–2 g of pastry sample (accurate to 0.0001 g) was weighed into a 50 mL plastic centrifuge tube, 100 μL of internal standard solution D4-hydroxybenzoic acid ethyl ester (1 mg/mL) was added, and 0.5 mL of hydrochloric acid (1 + 1, *v*/*v*) was added for acidification. This was extracted with 10–15 mL of acetonitrile, vortexed, and centrifuged, and 5 mL of the upper acetonitrile extract was taken out into a 15 mL plastic centrifuge tube. Then, 2 mL of n-hexane was added for defatting, the upper layer of n-hexane was discarded, 2 mL of saturated sodium chloride solution was added for impurity removal, vortexed, and centrifuged. The acetonitrile was extracted and concentrated to near dryness under nitrogen, and 1 mL of methanol was taken for vortexing and resolubilization, which was then analyzed.

### 2.6. Test of Recovery by Adding Standard

Blank samples, which did not contain the 12 preservatives, were chosen. Approximately 1–2 g (precisely to 0.0001 g) of this sample was weighed into a 50 mL plastic centrifuge tube. Then, 100 μL of the internal standard solution (1 mg/mL) D4-p-hydroxybenzoic acid ethyl ester was added. Subsequently, three different concentration levels of the mixed standard solution were separately added to achieve target compound contents of 10 mg/kg, 40 mg/kg, and 100 mg/kg in the sample. The samples were then processed according to the method described in Section 2.5, and the recovery rates were calculated. Each addition level was repeated in six parallel measurements to assess the accuracy and precision of the method.

## 3. Results and Discussion

### 3.1. Optimization of Detection Conditions

#### 3.1.1. Selection of Internal Standard

Due to the complex composition of the food matrix and the relatively low content of target compounds, they are easily interfered and masked, so a series of methods such as purification, enrichment, and concentration are needed for pretreatment. Some preservatives, such as acetic acid, propionic acid, and dimethyl fumarate, have small molecular weights, low boiling points, and are easy to sublime, which may cause the loss of target compounds during pretreatment, affecting the recovery rate. Therefore, it is necessary to select appropriate internal standard substances.

In this study, D4-hydroxybenzoic acid ethyl ester was selected as the internal standard for the determination of 12 preservatives, including acetic acid, propionic acid, benzoic acid, sorbic acid, and nipagin esters. Because D4-hydroxybenzoic acid ethyl ester has similar physical and chemical properties to the target compounds, it can dissolve completely in the tested sample, does not exist in the sample, and does not react with the tested sample. At the same time, it can separate chromatographic peaks of each component in the sample, and the peak position is close but not overlapped, which can effectively avoid the sensitivity difference caused by instrument instability.

#### 3.1.2. Selection of Characteristic Ions

For optimizing the detection conditions, it was crucial to carefully select the characteristic ions that have high peak intensity, minimal interference, and good matching degree. To this end, we used the full scan mode to measure 12 target compounds at a concentration of 1 mg/mL and selected the characteristic ions from various ion fragments. These ions were used for qualitative and quantitative analysis, and their retention times were roughly determined. To further improve method selectivity and sensitivity, we employed the selected ion monitoring (SIM) mode. The chromatograms, retention times, and characteristic ions of the 12 food preservatives and the internal standard are presented in Figure 1 and Table 2, respectively.

#### 3.1.3. Selection of Chromatographic Column

Under the same chromatographic mass spectrometry conditions, a DB-FFAP (30 m × 250 μm × 0.25 μm) chromatographic column and a VF-WAXms (30 m × 250 μm × 0.25 μm) chromatographic column were used to measure the same concentration of standard solution. The results showed that on the VF-WAXms chromatographic column, the baseline of the chromatogram was uneven, the peaks of the target compounds were relatively poor, and there was a tailing phenomenon. On the other hand, on the DB-FFAP chromatographic column, the separation of each target compound was good, the peak shape was good, and the response value was high. Therefore, in this experiment, a DB-FFAP chromatographic column was used to determine the target preservatives.

### 3.2. Optimization of Sample Preparation Conditions

#### 3.2.1. Selection of Extraction Solvents

Under the same conditions of other sample preparation and detection, this experiment used acetonitrile, acetone, methanol, water, ethanol, dichloromethane, ethyl acetate, petroleum ether-ether (3 + 1, *v*/*v*), and n-hexane-ethyl acetate (1 + 1, *v*/*v*) as extraction solvents to compare their extraction efficiencies. The results showed that the extraction efficiencies of acetic acid, propionic acid, and dimethyl fumarate were similar in all extraction solvents, while the extraction efficiency of dehydroacetic acid was relatively higher when extracted with ethyl acetate or acetonitrile, and the tailing of chromatographic peaks was less pronounced. Ether and n-hexane were highly volatile and unstable. Water had a significant impact on acetic acid and propionic acid, leading to poor extraction efficiency. Taking into account safety and extraction efficiency, acetonitrile was finally selected as the extraction solvent.

Because acetic acid, propionic acid, dehydroacetic acid, and other substances are prone to volatilization and have unstable properties, their extraction efficiency is low. This experiment used acetonitrile–water solutions of different proportions (3:1, 4:1, 5:1, 8:3, 10:3, and 10:0) for extraction. The extraction solution was dehydrated with anhydrous sodium sulfate. The results showed that the target compounds were lost to a large extent during the dehydration process and the extraction efficiency did not increase significantly. Therefore, acetonitrile was selected as the extraction solvent. Meanwhile, this experiment compared the effects of directly injecting the acetonitrile extraction solution and concentrating it with nitrogen blowing and then resuspending it with methanol before injection. It was found that the peak shape of acetic acid, propionic acid, and dehydroacetic acid was more stable when resuspended in methanol. Therefore, this experiment chose to resuspend the acetonitrile extraction solution with methanol before injection.

#### 3.2.2. Purification Conditions

Pastry samples usually contain fats and sugars and require appropriate reagents and methods to purify the samples by removing fats and impurities. In this experiment, 2 mL of n-hexane was used for different frequencies (one, two, and three times) of defatting experiments, and the results showed that one round of vortex defatting with 2 mL of n-hexane was sufficient to meet the experimental requirements. Additionally, 2 mL of saturated sodium chloride solution was used for different frequencies (one, two, and three times) of impurity removal experiments, and the results showed that one round of vortex impurity removal with 2 mL of saturated sodium chloride solution was sufficient to meet the experimental requirements. Moreover, high-speed and long-time vortex and centrifugation can easily cause emulsification interference in the experiment. After multiple experimental adjustments, it was found that a rotation speed of 5000 r/min and a time of 3 min could achieve good vortex and centrifugation effects without causing sample emulsification. The sample determination chromatogram is shown in Figure 2.

### 3.3. Methodological Validation

#### 3.3.1. Linearity, Detection Limit, and Quantification Limit

Under the final determined measurement conditions of this method, the linearity, detection limit, and quantification limit demonstrate that within the range of 1.0–50 μg/mL, 12 preservatives exhibit good linearity, with correlation coefficients greater than 0.99. The detection limit of the method was calculated using the instrument’s 3-fold signal-to-noise ratio, resulting in a range of 0.04–2.00 mg/kg. The quantification limit was validated, and the results are presented in Table 3. The quantification limit was calculated using the instrument’s 10-fold signal-to-noise ratio, resulting in a range of 0.12–6.67 mg/kg.

#### 3.3.2. Accuracy and Precision

According to the established method, blank samples without the 12 preservatives were selected for the addition experiment at three different concentration levels: low, medium, and high. Each level was repeated six times for parallel determination to evaluate the accuracy and precision of the method. The results are shown in Table 4 and Table 5. The results indicate that the average recovery rate of the 12 preservatives is between 70.18% and 109.22%, and the relative standard deviation is between 1.82% and 9.79%. The method is accurate, reliable, and suitable for the analysis and research of 12 preservatives, such as benzoic acid, sorbic acid, and nipagin esters, in pastry foods.

### 3.4. Actual Sample Analysis

Using the method established in this experiment, preservative testing was conducted on pastry foods such as puff pastry biscuits, soda biscuits, cakes, and toast, as shown in Table 6 and Table 7. The results showed that all samples tested positive for preservatives, with the highest detection rates for acetic acid, dehydroacetic acid, sorbic acid, and butylparaben. The detected amounts of acetic acid ranged from ND to 73.82 mg/kg, propionic acid from ND to 150.53 mg/kg, dimethyl fumarate from ND to 1.85 mg/kg, sorbic acid from ND to 540.56 mg/kg, dehydroacetic acid from ND to 272.68 mg/kg, benzoic acid from ND to 25.69 mg/kg, isopropylparaben from ND to 2.42 mg/kg, methylparaben from ND to 2.85 mg/kg, ethylparaben from ND to 2.94 mg/kg, propylparaben from ND to 2.71 mg/kg, and butylparaben was not detected. In GB2760-2014, there are no specific quantitative limits defined for acetic acid, dimethyl fumarate, and benzoic acid in pastries. However, the regulation stipulates the limits for propionic acid, dehydroacetic acid, sorbic acid, and methyl paraben in pastries to be 2.5 g/kg, 0.5 g/kg, 1.0 g/kg, and 0.5 g/kg, respectively. The detected values did not exceed the maximum allowable usage levels of the National Food Safety Standards for Food Additives (GB2760). Most of the samples tested positive for more than one preservative, and the total detected amount of preservatives complied with the regulations that when using multiple food additives with the same function, the ratio of each additive’s usage amount to its maximum usage amount does not exceed 1. However, in some samples, the actual detected preservatives were inconsistent with the label information. Utilizing this approach for testing commercially available pastries, a portion of positive samples were concurrently examined using national standard methods. The results obtained from this method align closely with those obtained through the national standard methods. In the literature, Arias, J.L.O. et al. employed the QuEChERS technique and HPLC-UV method to determine preservatives in various processed foods, analyzing benzoic acid, sorbic acid, and methyl paraben in 82 samples [30]. Wu Yi, Zhou Lujun, and Huang Cheng, among others, individually employed high-performance liquid chromatography to detect multiple additives [31,32,33]. However, these methods typically only encompass three to five types of preservatives. Their limitations lie in the restricted variety of preservatives tested, with certain preservatives such as propionic acid, acetic acid, and dimethyl fumarate being excluded. This study’s methodology addresses this gap, presenting a convenient, rapid, and precise multi-component detection method.

## 4. Conclusions

Overall, the study established a reliable and accurate method for the analysis of 12 preservatives in pastry foods using gas chromatography–mass spectrometry with internal standard calibration. The method was found to be simple to operate, rapid, sensitive, and precise, with the use of internal standard calibration minimizing errors caused by pre-experimental treatment. Additionally, the use of gas chromatography–mass spectrometry reduced false-positive errors through retention time and characteristic ion identification. The results showed that most of the pastry food samples contained at least one type of preservative, but the detected levels were all within the maximum allowable usage limits according to the national food safety standards. The study’s findings indicate that the established method is suitable for the fast and effective detection of preservatives in pastry foods.

## Figures and Tables

**Figure 1 foods-12-03819-f001:**
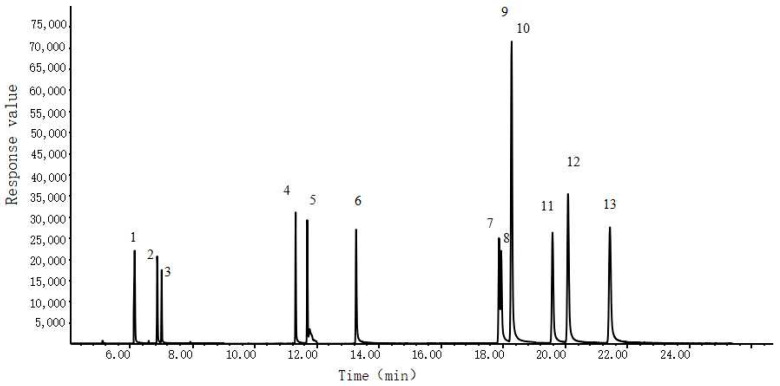
Chromatogram of 12 preservatives and their internal standards. Peak identifications: 1. acetic acid; 2. propionic acid; 3. dimethyl fumarate; 4. sorbic acid; 5. dehydroacetic acid; 6. benzoic acid; 7. isopropyl p-hydroxybenzoate; 8. methyl p-hydroxybenzoate; 9. ethyl p-hydroxybenzoate; 10. D4-ethyl p-hydroxybenzoate; 11. propyl p-hydroxybenzoate; 12. isobutyl p-hydroxybenzoate; 13. butyl p-hydroxybenzoate.

**Figure 2 foods-12-03819-f002:**
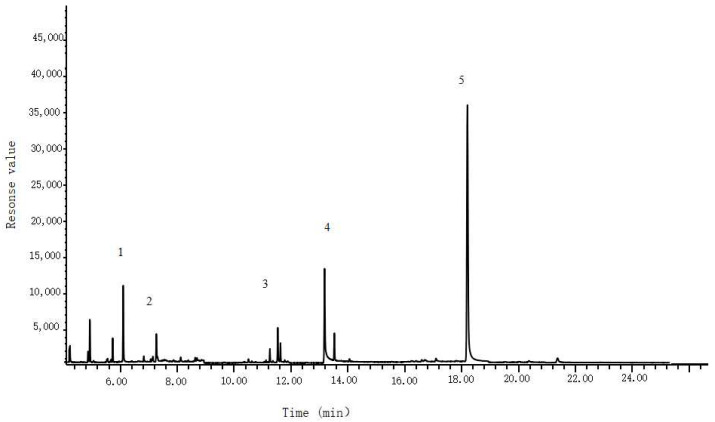
Chromatogram of 12 preservatives for the determination of scallion flavor soda cake. Peak identifications: 1. acetic acid; 2. Ppropionic acid; 3. dehydroacetic acid; 4. benzoic acid; 5. d4-ethyl p-hydroxybenzoate.

**Table 1 foods-12-03819-t001:** Corresponding standard series concentrations of target compounds.

Compound Name	Standard Series Concentration (μg/mL)				
Acetic acid, propionic acid, sorbic acid, benzoic acid, dehydroacetic acid	50	25	12.5	5	2.5
Dimethyl fumarate, methyl p-hydroxybenzoate, isobutyl ester, butyl ester, Ethyl p-hydroxybenzoate, propyl ester, Isopropyl p-hydroxybenzoate	20	10	4	2	1

**Table 2 foods-12-03819-t002:** Retention time and quantitative and qualitative selection ions of 12 preservatives.

Food Preservatives	Quantitative Ion (*m*/*z*)	Qualitative Ion (*m*/*z*)		Retention Time (min)
Acetic acid	60	43	45	6.161
Propionic acid	74	73	57	6.875
Dimethyl fumarate	113	85	59	7.011
Sorbic acid	112	97	67	11.342
Dehydroacetic acid	168	153	85	11.626
Benzoic acid	105	122	77	13.280
Isopropyl p-hydroxybenzoate	121	138	180	17.888
Methyl p-hydroxybenzoate	121	152	93	17.967
D4-p-hydroxybenzoate	125	142	170	18.295
Ethyl p-hydroxybenzoate	121	138	166	18.296
Propyl p-hydroxybenzoate	121	138	180	19.618
Isobutyl p-hydroxybenzoate	121	138	65	20.113
Butyl p-hydroxybenzoate	121	138	194	21.469

This is example 1.

**Table 3 foods-12-03819-t003:** Linear range, linear equation, correlation coefficient, detection limit, and quantitative limit of 12 preservatives.

Food Preservatives	Linear Range (μg/mL)	Regression Equation	r²	Detection Limit/(mg/kg)	Quantitative Limit/(mg/kg)
Acetic acid	2.5–50	y = 5157x − 1600	0.9967	0.17	0.58
Propionic acid	2.5–50	y = 7128x − 0900	0.9993	0.60	1.99
Dimethyl fumarate	1.0–20	y = 19,160x + 10,680	0.9942	0.28	0.92
Sorbic acid	2.5–50	y = 3089x − 263	0.9923	1.83	6.10
Dehydroacetic acid	2.5–50	y = 4613x − 451	0.9904	2.00	6.67
Benzoic acid	2.5–50	y = 8941x − 9780	0.9924	1.38	4.62
Isopropyl p-hydroxybenzoate	1.0–20	y = 31,090x − 634	0.9993	0.04	0.14
Methyl p-hydroxybenzoate	1.0–20	y = 31,470x + 961.2	0.9973	0.04	0.12
Ethyl p-hydroxybenzoate	1.0–20	y = 32,140x − 53	0.9979	0.04	0.12
Propyl p-hydroxybenzoate	1.0–20	y = 30,030x + 669	0.996	0.11	0.36
Isobutyl p-hydroxybenzoate	1.0–20	y = 32,310x + 18780	0.99	0.33	1.03
Butyl p-hydroxybenzoate	1.0–20	y = 27,050x − 024	0.9974	0.05	0.18

**Table 4 foods-12-03819-t004:** Accuracy and precision of 5 food preservatives (n = 6) %.

Food Preservatives	Low Concentration (10 mg/kg)		Medium Concentration (40 mg/kg)		High Concentration (100 mg/kg)	
	Average Recovery Rate	Relative Standard Deviation	Average Recovery Rate	Relative Standard Deviation	Average Recovery Rate	Relative Standard Deviation
Acetic acid	75.85	2.37	71.19	4.08	70.18	4.05
Propionic acid	79.6	4.72	77.93	2.78	70.29	3.89
Sorbic acid	90.28	2.27	81.48	3.12	80.73	1.82
Dehydroacetic acid	78.6	3.48	76.07	4.29	73.72	6.28
Denzoic acid	86.01	3.39	83.39	2.75	81.73	5.42

**Table 5 foods-12-03819-t005:** Accuracy and precision of 7 food preservatives (n = 6) %.

Food Preservatives	Low Concentration (4 mg/kg)		Medium Concentration (20 mg/k)		High Concentration (40 mg/kg)	
	Average Recovery Rate	Relative Standard Deviation	Average Recovery Rate	Relative Standard Deviation	Average Recovery Rate	Relative Standard Deviation
Dimethyl fumarate	107.97	5.28	98.45	2.69	85.24	9.27
Isopropyl p-hydroxybenzoate	98.32	8.44	104	7.26	109.22	2.97
Methyl p-hydroxybenzoate	103.79	8.98	102.91	6.72	100.21	9.79
Ethyl p-hydroxybenzoate	98.97	5.48	101.73	3.43	96.48	7.73
Propyl p-hydroxybenzoate	102.83	8.14	101.07	8.25	98.05	8.64
Isobutyl p-hydroxybenzoate	81.82	2.85	83.87	4.3	83.28	3.91
Butyl p-hydroxybenzoate	106.72	6.27	107.05	5.81	87.4	9.11

**Table 6 foods-12-03819-t006:** Detection results of acetic acid, propionic acid, dimethyl fumarate, sorbic acid, dehydroacetic acid, benzoic acid in pastry samples (mg/kg).

	Acetic Acid	Propionic Acid	Dimethyl Fumarate	Sorbic Acid	Dehydroacetic Acid	Benzoic Acid
Scallion-flavored Soda Cake	22.23	1.02	ND	ND	7.39	25.69
Rice noodles	10.92	ND	ND	ND	ND	ND
Osmanthus Lotus Root Cake	14.19	ND	ND	30.34	ND	ND
Corn flour	ND	ND	ND	ND	ND	4.39
Pacific Soda Cake	ND	ND	ND	6.23	ND	4.39
Evergreen Crispy Biscuit	ND	ND	1.85	ND	ND	ND
Sugar-free, high-fiber crispy biscuits	ND	ND	ND	ND	ND	ND
Gangrong Steamed Cake	6.88	ND	ND	520.63	114.84	14.52
Gangrong Steamed Bread	73.82	82.48	ND	68.03	175.15	4.51
Gangrong Sandwich Sandwich Cake	3.64	ND	ND	439.19	92.48	ND
Barbie Bear Pure Cake	11.20	1.88	ND	267.05	103.93	ND
Barbie Bear Fresh Cake	18.95	1.87	ND	269.20	120.36	ND
Hometown Sweet Pork Floss and Cheese Small Rolls	13.13	ND	ND	540.56	149.02	ND
Hometown Fruit Cake	4.31	ND	ND	537.69	171.81	ND
Hamburg cake	17.26	ND	ND	347.71	127.62	ND
White Peach Milk Cake	14.41	ND	ND	381.25	90.32	ND
Cheese milk cake	22.35	ND	ND	439.79	102.59	ND
Sucrose-free cake	9.48	ND	ND	360.67	111.34	ND
Sea salt cheese cake	40.66	2.29	ND	346.02	105.75	3.11
Milk vanilla-flavored small cake	4.14	ND	ND	539.48	74.59	ND
Honey cheese-flavored small cake	8.91	ND	ND	471.85	77.00	ND
Condensed milk toast	73.79	150.53	ND	72.75	207.95	2.93
Purple Rice Toast	73.35	137.54	ND	147.31	272.68	ND
Cheese toast	72.27	142.08	ND	80.96	217.06	ND
Yeast Salty Toast	66.03	122.34	ND	137.45	225.30	ND

**Table 7 foods-12-03819-t007:** Detection results of isopropyl p-hydroxybenzoate, methyl p-hydroxybenzoate, ethyl p-hydroxybenzoate, propyl p-hydroxybenzoate, propyl p-hydroxybenzoate, isobutyl p-hydroxybenzoate, butyl p-hydroxybenzoate in pastry samples (mg/kg).

	Isopropyl P-Hydroxybenzoate	Methyl P-Hydroxybenzoate	Ethyl P-Hydroxybenzoate	Propyl P-Hydroxybenzoate	Isobutyl P-Hydroxybenzoate	Butyl P-Hydroxybenzoate
Scallion-flavored Soda Cake	ND	ND	ND	ND	ND	ND
Rice noodles	ND	ND	ND	ND	ND	ND
Osmanthus Lotus Root Cake	ND	ND	ND	ND	ND	ND
Corn flour	0.63	1.91	2.23	2.29	ND	ND
Pacific Soda Cake	ND	ND	ND	ND	ND	ND
Evergreen Crispy Biscuit	2.42	2.85	2.94	2.71	ND	ND
Sugar-free, high-fiber crispy biscuits	2.40	1.92	1.90	1.76	ND	ND
Gangrong Steamed Cake	ND	ND	ND	ND	ND	ND
Gangrong Steamed Bread	ND	ND	ND	ND	ND	ND
Gangrong Sandwich Sandwich Cake	ND	ND	ND	ND	ND	ND
Barbie Bear Pure Cake	ND	ND	ND	ND	ND	ND
Barbie Bear Fresh Cake	ND	ND	ND	ND	ND	ND
Hometown Sweet Pork Floss and Cheese Small Rolls	ND	ND	ND	ND	ND	ND
Hometown Fruit Cake	ND	ND	ND	ND	ND	ND
Hamburg cake	ND	ND	ND	ND	ND	ND
White Peach Milk Cake	ND	ND	ND	ND	ND	ND
Cheese milk cake	ND	ND	ND	ND	ND	ND
Sucrose-free cake	ND	ND	ND	ND	ND	ND
Sea salt cheese cake	ND	ND	ND	ND	ND	ND
Milk vanilla-flavored small cake	ND	ND	ND	ND	ND	ND
Honey cheese-flavored small cake	ND	ND	ND	ND	ND	ND
Condensed milk toast	ND	ND	ND	ND	ND	ND
Purple Rice Toast	ND	ND	ND	ND	ND	ND
Cheese toast	ND	ND	ND	ND	ND	ND
Yeast Salty Toast	ND	ND	ND	ND	ND	ND

## Data Availability

The data presented in this study are available on request from the corresponding author.

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
