# Peer review of "Simultaneous Determination of 12 Preservatives in Pastries Using Gas Chromatography–Mass Spectrometry"

_foods, 2023, doi:10.3390/foods12203819_

Round 1

Reviewer 1 Report

In the abstract, the background of the study is not justified. It should be told about the need for the study.

How the current technique is better than the others, explain it briefly in the introduction.

The details of parameters used for method validation are missing in the material and method section though; in short, it is mentioned in the results section. It would be great to include details in the material and method section.

Organize the manuscript carefully.

The discussion section can be improved by adding a similar study and comparing it with the current one. 

Section 2.3. Preparation of Solutions, 2.4. Preparation of Standard Curve and 22.5. Sample preparation (check some other parts also) must be in the past tense as the work has been already performed. There are many syntax errors and grammatical mistakes in the manuscript. There is a need to improve the English of the manuscript.

Some references are not as per journal guidelines. 

There are many syntax errors and grammatical mistakes in the manuscript. There is a need to improve the English of the manuscript.

Author Response

Dear reviewer,

Thank you very much! I have made revisions based on your feedback. Below, the text in red font represents my responses to each issue. For detailed modifications, please refer to the manuscript. Thank you.

Reviewer 2 Report

Foods

foods-2634266

Simultaneous determination of 12 preservatives in pastries using gas chromatography-mass spectrometry

Dear Editor,

The article deals with the establishment of a gas chromatography-mass spectrometry (GC-MS) method for the simultaneous determination of 12 preservatives in pastry. The topic is good. However, it needs major revision . Some specific points;

-       Line 28: Are all food preservatives antimicrobial agents?

-       Please highlight the aim of the study in the introduction section!

-       Lines 88 and 103: What is D4? Check the standard name! Is that deuterium? Do you have to use deuterium standards in the process?

-       Line 104: 10 or 15 ml?

-       Explain why some recoveries are higher than 100%!

-       Please give the LOD and LOQ values of the individual components!

-       Table 6: The results are corrected or uncorrected?

-       Line 232: Please give the individual limits of the compounds!

-       There is slippage in the text of the article

-       Please discuss the presence/amounts of these preservatives in the literature!

-       Discussion sections should be improved!

It is fine.

Author Response

(The authors gave the same response as above.)

Reviewer 3 Report

This article deals with method development and validation for determination of 12 common pastry preservatives using GC-MS, intended for product quality and safety control, thus corresponding with the objectives of the Special Issue.

Despite interesting article`s subject and acceptable research outcome results, the manuscript requires more editing (in this state it is also hard to read).

The text needs some more adjustment, e.g. font size and type (difference between first part of Introduction up to line 39 and the rest of it; Table 1 font size and type, also it breaks across two pages – Table 2 as well, and others…), etc. Line 221, line 218, 206, 239, …? Not sure if the (proper) foods template was used for writing.

Line 56-57: 1. Materials and Methods and then 2.1. Instruments and reagents…?

Instruments and reagents: Formula writing (index?), more data on equipment producers/providers.

Line 64: pure.All --- Space missing

Line 74: 1.0μL --- Space missing

Line 90: 1 mg/ml --- 1mg/mL, also elsewhere (e.g. Sample prep)

Line 80, preparation of solutions, standard curve, sample prep: Should describe the preparation without stating it as instructions from SOPs.

Line 102, elsewhere: 1-2g, 10-15ml --- Space missing

Line 207, Accuracy and precision: Could state what were the threshold limits for the investigated method performance characteristics to say that the obtained results were satisfying? Any legislation requirements existing/followed?

Discussion: Paragraph present work of other authors on this subject is missing --- What did other authors attempted to do, what methods were developed so far, how this one differs, etc.

Minor English language editing required

Author Response

(The authors gave the same response as above.)

Round 2

Reviewer 2 Report

Good revision

Reviewer 3 Report

I have checked the revised manuscript and author`s comments. Although manuscript needs some additional editing (tables breaking across two pages, text in lines 236 or 248, etc.), authors conducted sufficient modifications and provided clarifications for the stated issues.